# Comparative LC-ESIMS-Based Metabolite Profiling of *Senna italica* with *Senna alexandrina* and Evaluating Their Hepatotoxicity

**DOI:** 10.3390/metabo13040559

**Published:** 2023-04-13

**Authors:** Elaheh Zibaee, Maryam Akaberi, Zahra Tayarani-Najaran, Karel Nesměrák, Martin Štícha, Naghmeh Shahraki, Behjat Javadi, Seyed Ahmad Emami

**Affiliations:** 1Department of Traditional Pharmacy, School of Pharmacy, Mashhad University of Medical Sciences, Mashhad 13131-99137, Iran; 2Department of Pharmacognosy, School of Pharmacy, Mashhad University of Medical Sciences, Mashhad 13131-99137, Iran; 3Targeted Drug Delivery Research Center, Pharmaceutical Technology Institute, Mashhad University of Medical Sciences, Mashhad 13131-99137, Iran; 4Department of Analytical Chemistry, Faculty of Science, Charles University, 128 43 Prague, Czech Republic; 5Mass Spectrometry Laboratory, Section of Chemistry, Faculty of Science, Charles University, 128 43 Prague, Czech Republic; 6Medical Toxicology Research Center, Pharmaceutical Technology Institute, Mashhad University of Medical Sciences, Mashhad 13131-99137, Iran

**Keywords:** *Senna italica*, *Senna alexandrina*, LC-ESIMS, LC-MS/MS, sennosides, hepatotoxicity

## Abstract

*Senna* Mill. (Fabaceae) is an important medicinal plant distributed worldwide. *Senna alexandrina* (*S. alexandrina*), the officinal species of the genus, is one of the most well-known herbal medicines traditionally used to treat constipation and digestive diseases. *Senna italica* (*S. italica*), another species of the genus, is native to an area ranging from Africa to the Indian subcontinent, including Iran. In Iran, this plant has been used traditionally as a laxative. However, very little phytochemical information and pharmacological reports investigating its safety of use are available. In the current study, we compared LC-ESIMS metabolite profiles of the methanol extract of *S. italica* with that of *S. alexandrina* and measured the content of sennosides A and B as the biomarkers in this genus. By this, we were able to examine the feasibility of using *S. italica* as a laxative agent like *S. alexandrina.* In addition, the hepatotoxicity of both species was evaluated against HepG2 cancer cell lines using HPLC-based activity profiling to localize the hepatotoxic components and evaluate their safety of use. Interestingly, the results showed that the phytochemical profiles of the plants were similar but with some differences, particularly in their relative contents. Glycosylated flavonoids, anthraquinones, dianthrones, benzochromenones, and benzophenones constituted the main components in both species. Nevertheless, some differences, particularly in the relative amount of some compounds, were observed. According to the LC-MS results, the amounts of sennoside A in *S. alexandrina* and *S. italica* were 1.85 ± 0.095% and 1.00 ± 0.38%, respectively. Moreover, the amounts of sennoside B in *S. alexandrina* and *S. italica* were 0.41 ± 0.12 % and 0.32 ± 0.17%, respectively. Furthermore, although both extracts showed significant hepatotoxicity at concentrations of 50 and 100 µg/mL, they were almost non-toxic at lower concentrations. Taken together, according to the results, the metabolite profiles of *S. italica* and *S. alexandrina* showed many compounds in common. However, further phytochemical, pharmacological, and clinical studies are necessary to examine the efficacy and safety of *S. italica* as a laxative agent.

## 1. Introduction

*Senna* Mill. is a widespread genus of flowering and perennial plants belonging to the family Fabaceae. This genus has about 282 accepted species distributed worldwide, mostly in tropical and subtropical regions (Table 1) [1,2,3]. *Senna alexandrina* (*S*. *alexandrina*) (Syn. *Cassia acutifolia* Delile; *Cassia alexandrina* (Mill.) Spreng.; *Cassia senna* L.; *Senna acutifolia* Batka), known as Indian senna, is one of the most well-known herbal medicines in the genus (Figure 1A) [4,5]. In different systems of traditional medicine, *S. alexandrina* has been mainly used as a laxative to relieve constipation [6,7]. In addition, *S. alexandrina* has been used traditionally for the treatment of epilepsy, bronchial congestion, typhoid, skin diseases, malaria, intestinal gas, yellow fever, dyspepsia, infections, cholera, and hemorrhoids [8,9,10,11,12,13]. Phytochemically, *S. alexandrina* comprises mostly anthraquinones (including sennosides A, B, C, and D; rhein; and aloe-emodin), which are potent natural laxatives. The sennosides A and B as senna biomarkers are optic isomers distinguishable on the configuration of the C-10 and C-10’. Moreover, the presence of similar anthraquinones and anthrones has been reported in other species of the genus such as *Senna tora* [14,15], *Senna alata* [16,17], and *Senna occidentalis* [18,19]. However, other phytochemicals including phenolic compounds (isorhamnetin, kaempferol, and kaempferin), glycosides, steroids, and saponins have been isolated from the plants belonging to this genus [10,20,21,22]. Several pharmaceutical activities such as antioxidant [23], anticancer [24], antidiabetic [25], anti-inflammatory [26,27], antimicrobial [28,29], and antihepatoma activity [30] have been reported for *S. alexandrina*.

*Senna italica* (*S. italica*) is native to countries from Africa to the Indian subcontinent, including Iran (Table 1, Figure 1B). In Iran, it is distributed in Hormozgan province and is known as Kowsen [31]. In Hormozgan province, the plant has been used as a laxative and cathartic agent [32]. There are a few studies investigating the specialized metabolites in *S. italica*. In these studies, compounds such as physcion, chrysophanol, 1,5-dihydroxy-3-methyl anthraquinone, 10,10’-chrysophanol bianthrone, chrysophanol-physcion bianthrone, and chrysophanolisophyscion bianthrone have been reported as the main anthraquinone derivatives in *S. italica* [33]. *Senna italica* has shown several pharmaceutical properties such as antioxidant [34], cytotoxicity [35], antibacterial [36,37], antiproliferative [36], hypoglycaemic [38], anti-obesity [38], and anti-inflammatory activities [39,40].

Hepatotoxicity is a major health concern in consuming many medicinal plants. As *S. alexandrina* is widely used as a laxative agent, some studies have investigated the hepatotoxicity of this valuable herbal medicine. On the basis of the literature review, *S. alexandrina* has been associated with hepatotoxicity [41]. In addition, the safety of sennosides has been evaluated in a wide range of toxicity experiments. Sennosides have been identified as the hepatotoxin anthraquinone glycosides present in *S. alexandrina*. Moreover, the overdose of laxatives containing anthraquinone glycosides (sennosides) and the long-term treatment of senna could increase the risk of colon cancer mediated through the overexpression of p53 and p21 [42]. Toyoda et al. reported that the regenerative processes that occurred after inflammatory or cytotoxic changes in response to laxatives (including *S. alexandrina*), and stimulation would be responsible for inducing cell proliferation. However, sennosides could function as an anti-tumor agent in a low dose. For instance, the administration of sennoside A (10 mg/kg in vivo and 10 μM in vitro) could inhibit the growth of pancreatic cancer cells [43]. To the best of our knowledge, there is no study investigating the hepatotoxicity and safety of *S. italica*.

In the current study, we aimed to compare the LC-ESIMS and LC-ESIMS/MS metabolite profiles of *S. italica* with *S. alexandrina.* By this technique, we were able to evaluate the similarities and differences in their major metabolites. Moreover, the contents of sennosides A and B, as the biomarkers in this genus, were measured in both species. In addition, the hepatotoxicity of *S. italica* and *S. alexandrina* were evaluated in an in vitro model of study.

## 2. Materials and Methods

### 2.1. Solvents and Chemicals

HPLC-grade solvents were obtained from Merck Millipore (Darmstadt, Germany). For extraction and preparative separation, technical-grade solvents (Mojallali, Iran) were used after distillation. Analytical-grade formic acid (FA), trifluoroacetic acid (TFA), and sulfuric acid were sourced from Merck Millipore (Darmstadt, Germany). DMSO was purchased from Sigma-Aldrich (St. Louis, MO, USA). Deionized water and ultrapure water were prepared using Elix and Milli-Q water purification systems (Merck Millipore, Darmstadt, Germany). Silica gel 60 F254-coated aluminum TLC plates were obtained from Merck (Darmstadt, Germany). AlamarBlue^®^ was purchased from Sigma (Saint Louis, MO, USA); RPMI-1640 and FBS were obtained from Gibco. The analytical standards of sennoside A and sennoside B were purchased from Sigma-Aldrich. Acetic acid (LC-MS purity), methanol (Hypergrade for LC-MS), and sodium acetate (99.995% trace metals basis) were purchased from Sigma-Aldrich.

A liquid chromatograph UHPLC Nexera XR (Shimadzu, Japan) connected with a Compact QTOF Bruker mass spectrometer (Bruker, Germany) using Compass Otofcontrol 4.0 software (Bruker Daltonics, Germany) was used for ESI-MS and ESI-MS/MS analysis of the extracts. Data processing was performed by Compass Data Analysis 4.4 software (Build 200.55.2969) (Bruker Daltonics, Germany). The MS data were collected in negative ionization mode at a scan range of *m*/*z* 50–1000. The temperature of the drying gas was 220 °C with a flow rate of 3 dm^3^/min. The cone voltage was 2800 V.

### 2.2. Plant Materials

*Senna italica* aerial parts were collected from Bandar Abbas, Hormozgān Province (Iran), in March 2018. The collected plants were dried at room temperature in the shade and stored in paper bags until extraction. *Senna alexandrina* was purchased from an herbal store in Mashhad. The plants were identified by botanist Mitra Suzani (Department of Pharmacognosy, School of Pharmacy, Mashhad University of Medical Sciences, Mashhad, Iran). Voucher specimens for *S. italica* (Voucher number 13576) and *S. alexandrina* have been deposited at the Herbarium of the Faculty of Pharmacy, Mashhad University of Medical Sciences (MUMS), Mashhad, Iran.

### 2.3. Extraction

To obtain 100% methanol (MeOH) extracts, the dried aerial parts of both plants (100 g) were milled and exhaustively percolated with MeOH (1 L, 24 h) [44]. The extracts were filtered and dried under reduced pressure by a rotary evaporator to afford 16.45 and 13.66 g of solid residue for *S. alexandrina* (16.45% *w*/*w*) and *S. italica* (13.66% *w*/*w*), respectively. To obtain 70% MeOH extracts, 25 g of each plant were percolated with 70% aqueous MeOH for 24 h. The solvents of the obtained extracts were removed by a rotary evaporator to afford 7.65 and 5.36 g of solid residue for *S. alexandrina* (30.60% *w*/*w*) and *S. italica* (21.44% *w*/*w*), respectively. The extracts were stored in a −20 °C freezer until use.

### 2.4. Microfractionation for Activity Profiling

HPLC-based activity profiling of the samples was performed according to a previously established protocol [44] with some modifications. The microfractionation was performed using a semi-preparative HPLC instrument. Isolation was performed at 22 °C on a HPLC instrument (Knauer, Germany), equipped with a vacuum degasser, Smartline pump 1000, Smartline photodiode-array (PDA) detector 2800 with a UV light source, and a manual injector with a 2 mL sample loop. A Eurospher II 100-5 C_18_ column (250 mm × 8 mm, 5 μm) (Knauer, Germany) was used for reversed-phase separations, and the chromatograms were obtained at 266 and 340 nm (UV_max_ of sennosides A and B). Data processing was performed by EZChrome Elite software (Agilent, Germany). The extracts of both plants *S. italica* and *S. alexandrina* (100 mg/mL in DMSO) were submitted to HPLC-PDA in four portions of 50 mg (500 μL). The mobile phase was H_2_O (A) and MeOH (B), both containing 0.05% trifluoroacetic acid (TFA), and the following gradient profile was used: 25% B isocratic (0–1 min), 25% → 65% B (1–30 min), 65% → 100% B (30–31 min), 100% B (31–45 min). The flow rate was 2 mL/min, and 5 min fractions (between min 5 and 45) were collected into test tubes. Corresponding microfractions from the four separations were combined. After drying the plate, microfractions were tested for cytotoxic activity. A sample containing both plant extracts (50 mg/mL) was submitted to HPLC-PDA as a control.

### 2.5. HPLC-ESI-QqTOFMS and HPLC-ESI-QqTOFMS/MS Analyzes

In order to have a better insight into the metabolic profile of the plants, HPLC-ESIMS analysis of both 70% aqueous and 100% MeOH extracts obtained from *S. italica* and *S. alexandrina* was performed. Therefore, using an optimized chromatography condition [45], the compounds in all extracts were separated by LC technique and identified by an ESIMS detector. About 50 mg of the samples were weighed accurately and dissolved in MeOH in a volumetric flask to a final volume of 50.00 mL (i.e., the final concentration was 1 mg/mL). The separation was performed on a column XBridge^®^ BEH C18 (150 × 3.0 mm i.d., particle size 2.5 μm; Waters) at the temperature of 40 °C. The binary mobile phase of MeOH (solvent A) and 0.2% aqueous acetic acid (solvent B) was used with a flow rate of 0.2 mL/min. The following gradient mode was employed: 0.0–17.5 min: 25–65% A, 17.5–18.0 min: 65–80% A, 18.0–20.0 min: 80% A, 20.0–20.5 min: 80–25% A, and 20.5–25.0 min: 25% A. The injected volume was 2 μL. ESI-MS detection was conducted on a Bruker QqTOF compact instrument operated using Compass otofControl 4.0 (Bruker Daltonics, Bremen, Germany) software. Compass DataAnalysis 4.4 (Build 200.55.2969) (Bruker Daltonics, Bremen, Germany) software was used for data processing. ESI-MS data were collected in positive and negative ion modes at a scan range from *m*/*z* 50 to *m*/*z* 1000. The temperature of the drying gas was set to 220 °C at a 3.0 dm^3^/min flow rate. The cone voltage was 2800 V.

### 2.6. Determination of Sennoside A and Sennoside B

Four samples SA1 (*S. alexandrina*, 100% MeOH extract), SA2 (*S. alexandrina*, 70% aqueous MeOH extract), SI1 (*S. italica*, 100% MeOH extract), and SI2 (*S. italica*, 70% aqueous MeOH extract) were analyzed quantitatively by LC-MS to determine the content of sennosides. Fifty milligrams of each sample was weighed accurately and dissolved in MeOH in a volumetric flask to a final volume of 50.00 mL. The determination was performed according to the procedure published in [45]. Quantification was based on a five-point calibration. Each sample was measured in triplicate. The samples were stored at −18 °C before analysis.

### 2.7. MZmine Preprocessing

Raw MS data files were converted into mzML files and processed using MZmine 3 [46]. Mass detection was carried out with a centroid mass detector, and a noise level was set for both MS and MS2 levels. The ADAP chromatogram builder parameters including minimum group size of scans, minimum group intensity threshold, minimum highest intensity, and *m*/*z* tolerance were set. The wavelets algorithm (XCMS) was used for the chromatogram deconvolution and its settings including S/N threshold, intensity window SN, minimum feature height, coefficient area threshold, peak duration range, and RT wavelet range were optimized. Chromatograms were deisotoped using the isotopic peaks grouper algorithm with an optimized *m*/*z* tolerance, RT tolerance, and maximum charge of 2, and the representative isotope used was the most intense. The features were aligned using the join aligner algorithm with an optimized *m*/*z* tolerance, weight for *m*/*z*, retention time tolerance, and weight for RT. Then, the features were gap-filled using a peak finder algorithm with an optimized intensity tolerance, *m*/*z* tolerance, retention time tolerance, and minimum data points. In the end, the data were exported as a CSV file.

### 2.8. Data Clustering

Data mining resulted in a total of 21 features automatically assembled in a matrix table in Excel. The clustering was performed according to a protocol found anywhere [47,48]. The web application called FreeClust was used for visualizing the features as heatmaps. This application provides an interactive and freely accessible environment for statistical evaluation and data visualization. Initially, all relevant chromatographic and spectroscopic data (retention time and molecular ions with the respective ion abundance) were automatically extracted and assembled in an Excel spreadsheet. The file was then loaded into the online web application FreeClust, and the ESIMS/MS data sets were clustered using sparse hierarchical clustering methods.

### 2.9. Identification of Compounds

Identification of the metabolites in both *Senna* species was accomplished by their UV–VIS spectra (220–600 nm); retention times relative to external standards; mass spectra (full scan and tandem MS); and comparison to our in-house database, dictionary of natural products database (CRC), and reference literature.

### 2.10. Cell Culture

The liver (Hep G2) and B16F10 cancer cell lines were obtained from Pasteur Institute (Tehran, Iran) and maintained in RPMI-1640 medium containing 10% (*v*/*v*) fetal bovine serum, 100 U/mL penicillin, and 100 μg/mL streptomycin at 37 °C in a humidified atmosphere of 5% CO_2_ and 95% air.

### 2.11. Cell Viability

AlamarBlue^®^ assay was performed to assess the toxicity of the extracts against HepG2 and B16F10 cancer cell lines [49]. To screen cell viabilities, both cancer cell lines (1 × 10^4^ cells per well) were seeded in each well of a 96-well cell-culture plate in a total volume of 100 μL and incubated at 5% CO_2_ at a temperature of 37 °C for 24 h. Stock solutions (50 mg/mL) of the extract samples were prepared in DMSO and diluted (100, 50, 25, 12.5, 6.25, and 3.125 µg/mL) with the culture medium to ensure that the concentration of DMSO was less than 1% in all samples. Then, the cells were treated with the prepared concentrations of the samples and incubated for 48 h. Following this, resazurin reagent (20 μL) was added to each well, and the absorbance was assessed after 4 h of incubation, at 570 nm and 600 nm using an ELISA microplate reader (Awareness, Palm City, FL, USA) (23). The cytotoxicity of the extracts was expressed as IC_50_, calculated using Prism 5 Software (GraphPad, La Jolla, CA, USA), and presented as mean ± SEM of three independent experiments with three replicates for each concentration of tested extracts. For each study, a control sample remained untreated and received only medium in place of the test materials. Doxorubicin was used as a positive control at a concentration of 10 µg/mL. The cytotoxicity of the microfractions was also carried out according to the above-mentioned procedure, except that their cytotoxicity was evaluated only at concentrations of 50 and 100 µg/mL.

## 3. Results

### 3.1. Cytotoxicity

To evaluate the safety of *S. italica* and *S. alexandrina*, the cytotoxic activity of the 100% MeOH fractions was examined against HepG2 cells. Figure 2 shows the cell viability pattern in HepG2 cells after treatment with 100% MeOH extracts of *S. alexandrina* and *S. italica* in different concentrations (100–3.1 µg/mL). While both extracts showed significant cytotoxicity at concentrations 100 and 50 µg/mL, they were almost non-toxic at lower concentrations. In addition, the viability pattern for both extracts were similar.

The cytotoxic compounds in the 100% MeOH extracts of *S. italica* and *S. alexandrina* were tracked with the aid of HPLC-based activity profiling [44]. An activity profile of eight 5 min fractions and the corresponding LC−UV traces for *S. italica* and *S. alexandrina* are shown in Figure 3A,B, respectively. The cytotoxicity of the microfractions was evaluated at a concentration of 50 µg/mL. In general, the microfractions from *S. italica* were less cytotoxic than those from *S. alexandrina*. Major cytotoxicity for *S. italca* was observed for the time window between 25 and 30 min, while for *S. alexandrina,* the active time window was between 15–20 and 25–30 min (Figure 3).

### 3.2. HPLC-ESIMS and MS/MS Analysis

#### 3.2.1. Determination of Sennosides A and B

According to the LC-MS results, the amount of sennosides A and B were higher in *S. alexandrina* than *S. italica* samples (Table 2).

#### 3.2.2. LC-ESIMS and LC-ESIMS/MS Profiling

Figure 4 and Figure 5 show the LC-MS chromatograms of samples SA1, SA2, SI1, and SI2. As is obvious from Figure 4A,B, the constituents of *S. alexandrina* and *S. italica* appeared mostly in retention times between 15 and 20 min. In addition, Figure 5A,B shows that the relative quantity of compounds in 100% MeOH extracts in both plants was higher than that of 70% extracts.

The metabolite profiles of all samples (SA1, SA2, SI1, and SI2) were compared (Figure 6). Interestingly, the 100% and 70% MeOH extracts of *S. alexandrina* and *S. italica* were grouped into two distinct clusters. This shows that although some differences were observed for 70% and 100% extracts of each plant species, this difference was not significant. In general, *S. alexandrina* and *S. italica* were characterized by a rather similar metabolite pattern and were thus classified into two similar clusters (both in a cluster with blue color). Many of the components were common in both species.

#### 3.2.3. Identification of Compounds

A representative chromatogram (*Senna* preparation SP2) is presented in Figure 7. Metabolites were identified on the basis of their retention times, UV characteristics, and observed molecular and fragment ions (Table 3 and Table 4). The LC-ESIMS data mining resulted in fifteen components for each plant. The identified metabolites constituted various classes of compounds, including flavonoids, anthraquinones, and acetophenones. However, flavonoid glycosides were the major compounds in both species. Sennosides A (peaks 4 and 6) and B (peaks 10) were detected in both species (Figure 7 and Figure 8). ESIMS^−^ and ESIMS/MS of sennoside A/B are presented in Figure 9. A plausible ESI^−^-MS*^n^* fragmentation of sennoside A/B is illustrated in Figure 10 according to the observed precursor and product ions and literature data [45].

Sennosides A and B, rutin, 2-hydroxyemodin glucoside, and tinnevellin-O-glucoside/torachrysone O-glucoside/isotorachrysone O-glucoside were the common major components in both plant species. Peak 5 in *S. alexandrina* and peak 8 in *S. italica* were identified as rutin, which was a major compound in both species. Peaks 11 and 12 were identified as kaempferol-O-hexoside-pentoside and 2-hydroxyemodin glucoside, respectively. While in *S. italica*, the relative amount of peak 11 was more than *S. alexandrina*, the relative amount of peak 12 was higher in the latter. Peak 13 in both plants was identified as an O-glucoside derivative of tinnevellin, torachrysone, or isotorachrysone.

According to the results, it seems that flavonoids are more abundant in *S. alexandrina*. In addition, other classes of compounds such as anthraquinones were detected more in *S. italica* than *S. alexandrina*. For instance, whereas anthraquinones such as aloe-emodin and rhein as well as benzochromenones including nrrubrofusarin and toralactone were detected as major compounds in *S. italica*, they were absent in *S. alexandrina*.

## 4. Discussion

In an effort to compare the metabolites of *S. italica* with *S. alexandrina*, herein, we presented an LCESI/MS-based approach for metabolite profiling and relative phyto-equivalency evaluation of the two species. A broad spectrum of glycosides was identified. In both species, quercetin, isoquercetin, kaempferol, and rhamnetin/isorhamnetin glycosides were identified as the major flavonoids. Nevertheless, flavonoid derivatives were more abundant in *S. alexandrina* than *S. italica.* From the anthraquinone class of compounds, only 2-hydroxyemodin glucoside was detected in *S. alexandrina* as one of the major compounds, whereas aloe-emodin, rhein, and 2-hydroxyemodin were detected in *S. italica*. Naphthols including torachrysone/isotorachrysone and their glucoside were detected in both plants. Moreover, both main senna dianthrone glycosides, sennosides A and B, were detected and measured in both plants. According to the literature, the total content of sennosides A and B in dried senna leaflets varies in the range of 1.5–3.0%. Our findings demonstrated that sennosides A and B levels were higher in *S. alexandrina* than in *S. italica*. The content of sennoside A in *S. alexandrina* was nearly double that of *S. italica*. The amounts of sennoside A in *S. alexandrina* and *S. italica* were 1.85 ± 0.095% and 1.00 ± 0.38%, respectively. Moreover, the amounts of sennoside B in *S. alexandrina* and *S. italica* were 0.41 ± 0.12% and 0.32 ± 0.17%, respectively. Other sennosides including sennosides C and D were also detected in both species as minor constituents.

Studies show that more than 110 metabolites have been identified from the leaves and pods of *S. alexandrina* so far. While the reported metabolites mostly belong to flavonoids, viz., kaempferol, other classes of compounds including anthraquinones, bianthrones, and naphthols have been reported. As an example, Farag et al. identified the phytochemical constituents of different samples of *S. alexandrina* by LC-MS-based metabolomics [50]. They reported that glycosylated flavonoids such as kaempferol glycosides and isorhamnetin glycosides, anthraquinones aloe-emodin, emodin, and rhein (both in free and glycosylated forms); bianthrones sennosides A–D; and naphthols including torachrysone/isotorachrysone and their glycosides constituted the major compounds. Similarly, we observed that glycosylated flavonoids kaempferol, rhamnetin/isorhamnetin, and quercetin; bianthrones sennosieds A and B; and naphthols including torachrysone/isotorachrysone and their glycosides were the major components. However, in opposition to the study of Farag et al., in our study, 2-hyroxyemodin glucoside was the main anthraquinone derivative instead of aloe-emodin, emodin, and rhein. Compared to *S. alexandrina*, fewer phytochemical studies have been established on *S. italica,* so less phytochemicals have been reported so far. However, the same category of specialized metabolites including flavonoids, anthraquinones, bianthrones, and naphthols have been reported. For instance, Khalaf et al. isolated and identified physcion, emodin, 2-methoxy-emodin-6-O-β-D-glucopyranoside, tinnevellin, rutin, and 1,6,8-trihydroxy-3-methoxy-9,10-dioxo-9,10-dihydroanthracene from the ethyl acetate and *n*-butanol fractions of *S. italica* by NMR technique [55]. Recently, Omer et al. investigated the phenolic profile of *S. alexandrina* and *S. italica* by LC-MS profiling [51]. They found that rutin was the major compound in both *S. italica* (17,285.02 µg/g) and *S. alexandrina* (6381.85 µg/g). Hyperoside (SI: 1293.52 µg/g, SA: 3299.95 µg/g), quercetin (SI: 1207.88 µg/g), kaempferol-3-glucoside (SI: 1082.75 µg/g, SA: 3299.95 µg/g), isoquercitrin (SI: 1042.35 µg/g, SA: 2752.63 µg/g), and isorhamnetin (SI: 411.96 µg/g, SA: 454.4 µg/g) were the other major flavonoids in the plants. In our study, rutin was similarly one of the major flavonoids in both species. After that, while quercetin/isoquercetin, kaempferol, and rhamnetin/isorhamnetin were the main flavonoid derivatives in *S. alexandrina*, kaempferol and rhamnetin/isorhamnetin dominated in *S. italica*.

*Senna alexandrina* appears to have been medically used as a purgative to treat constipation since the ninth or tenth century. The anthraquinone derivatives particularly dianthrone glycosides in senna are responsible for the laxative effect [56,57,58]. Senna dianthrone glycosides, also known as sennosides, are medications commonly used to treat constipation and empty the large intestine before surgery [56,58]. In 2020, sennoside glycosides were the 291st most commonly prescribed drug in the United States under several brand names. The chief sennosides in senna are sennoside A and sennoside B. Sennoside A and B are optic isomers distinguishable on the configuration of the C-10 and C-10’ [45].

Like *S. alexandrina*, *S. italica* has been used as a laxative agent in folk and traditional medicine, particularly in Asian and African countries [59]. According to the literature and our findings, the chemical composition of *S. italica* is similar to that of *S. alexandrina*. However, the contents of dianthrones sennosides A and B in *S. italica* were less than *S. alexandrina*. Nevertheless, the number of identified anthraquinones in *S. italica* was higher than that of *S. alexandrina*; while aloe-emodin, rhein, and 2-hydroxyemodin glucosides were detected in *S. italica* as the major anthraquinones, only 2-hydroxyemodin glucoside was identified in *S. alexandrina* as the main anthraquinone derivative.

Studies show that sennosides are responsible for the laxative effects of senna. Mechanistically, sennosides act on and irritate the lining of the intestine wall, thereby causing increased intestinal muscle contractions leading to vigorous bowel movement. Sennoside A and B are metabolized into the active metabolite rhein anthrone by gut bacteria. Then, this metabolite can increase cyclooxegenase 2 (COX2) expression in macrophage cells, leading to an increase in prostaglandin E2 (PGE2). The increase in PGE2 can decrease the expression of aquaporin 3 (AQP3) in mucosal epithelial cells of the large intestine. The laxative effect is produced via the decrease in AQP3 expression that restricts water reabsorption by the large intestine, thereby increasing fecal water content. Moreover, rhein anthrone can stimulate peristalsis in the large intestine. In addition, studies show that rhein anthrone can excite submucosal acetylcholinergic neurons, resulting in increased secretion of chloride and prostaglandin. The movement of chloride ions into the large intestine would also help to draw water into the lumen [58,60].

In addition to senna dianthrones, studies show that senna anthraquinones also have laxative effects. As mentioned above, in the current study, the relative amount of anthraquinone derivatives was higher in *S. italica* than in *S. alexandrina*. In our study, rhein, aloe-emodin, emodin, and 2-hydroxy emodin glucosides were identified as the dominant anthraquinones in *S. italica*. In a mice model of study, rhein could improve motor function and colonic electromyography of constipation and reduce AQP3 expression in the colonic mucosa, thereby relieving the symptoms of constipation effectively [61]. Oppositely, Zhang et al. showed that long-term administration of rhein in Sprague-Dawley rats could develop the constipation via SCF/c-kit signaling pathway [62]. In another mice model of study but on emodin, the laxative effect of the compound has been attributed to an increase in AQP3 expression in the colon of mice [63]. Although laxative activities have been reported for senna anthraquinones, the main laxative effect of senna is due to the presence of sennosides [56,57,58].

## 5. Conclusions

Senna (*S. alexandrina*) is an important medicinal plant, particularly to treat constipation. *Senna alexandrina* is native to the Sahara and Sahel, as well as to the Indian subcontinent. Compared to *S. alexandrina*, *S. italica* has a wider distribution, from Africa to the Indian subcontinent. If the laxative properties of *S. italica* are proven, more plant resources can be used to produce plant-derived laxatives. Thus, the first step is evaluation of the phytochemical profile of *S. italica*. Our finding suggests that *S. italica* might be used as a laxative agent because it has a similar metabolite to *S. alexandrina*. However, since sennosides A and B are the main active constituents responsible for the laxative action of senna, *S. italica* might be used as a laxative agent with milder effects because it contains less amount of sennosides A and B. In addition, our findings showed that the plants were not cytotoxic at low concentrations. Nevertheless, further studies including pharmacological investigations and clinical trials are needed to evaluate the efficacy and safety of *S. italica* as a laxative.

## Figures and Tables

**Figure 1 metabolites-13-00559-f001:**
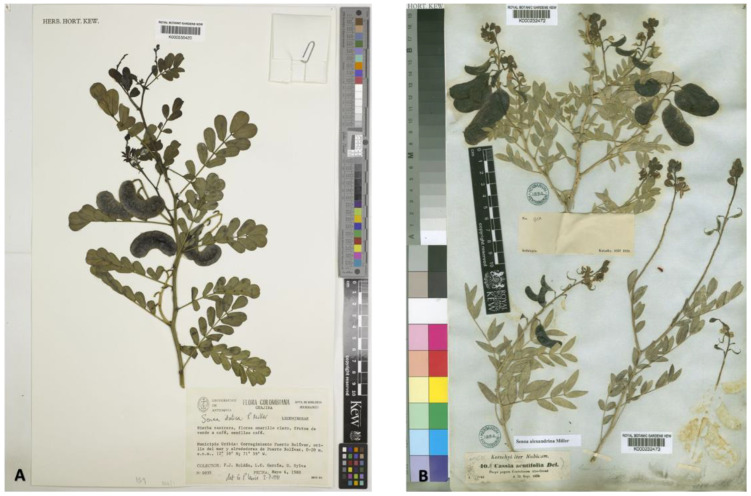
*Senna italica* (**A**) and *Senna alexandrina* (**B**) [5].

**Figure 2 metabolites-13-00559-f002:**
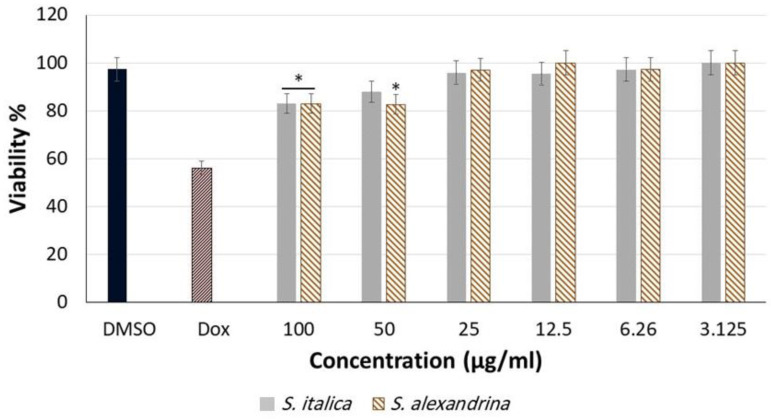
Viability of HepG2 cells after 48 h treatment with different concentrations (100–3.1 µg/mL) of the methanol extracts of *S. italica* and *S. alexandrina*. Doxorubicin (10 µg/mL) and DMSO (1%) were used as positive and negative controls, respectively. Analyses were carried out using the GraphPad Prism V.8 software and the one-way ANOVA test as well as Dunnett’s comparison test. The corresponding *p*-values marked with an asterisk are as follows: * < 0.05. The data are presented as mean ± SD (*n* = 3).

**Figure 3 metabolites-13-00559-f003:**
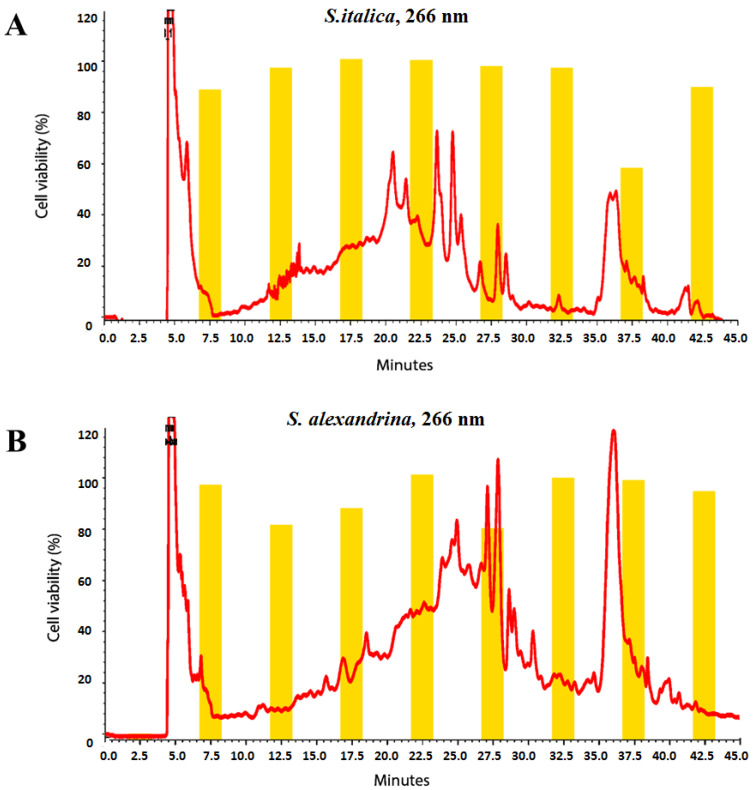
HPLC-based activity profiling of the 100% MeOH extract of (**A**) *Senna italica* and (**B**) *Senna alexandrina*. The UV chromatograms (266 nm) of separations of 50 mg of extracts on a semipreparative RP-HPLC column are shown. Activity of 5 min microfractions is indicated with colored columns for cytotoxic activity against HepG2 cells, expressed as % cell viability. The cytotoxicity of the microfractions were evaluated at concentration 50 µg/mL.

**Figure 4 metabolites-13-00559-f004:**
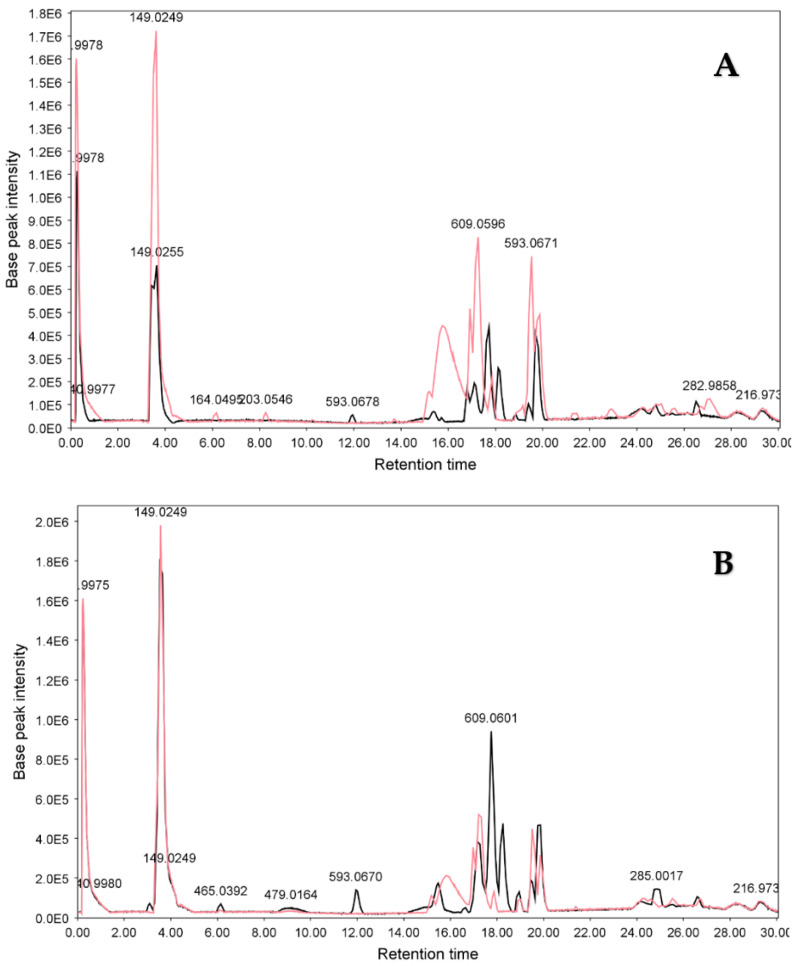
LC-MS chromatograms of (**A**) 100% and (**B**) 70% MeOH extracts of *S. alexandrina* (pink) and *S. italica* (black). For the separation, XBridge^®^ BEH C18 column (150 × 3.0 mm i.d., particle size 2.5 μm; Waters), gradient elution with MeOH and 0.2% aqueous acetic acid with a flow rate 0.2 cm^3^ min^−1^ was used. E5: 10^5^, E6: 10^6^. (For interpretation of the references to color in this figure legend, the reader is referred to the web version of this article).

**Figure 5 metabolites-13-00559-f005:**
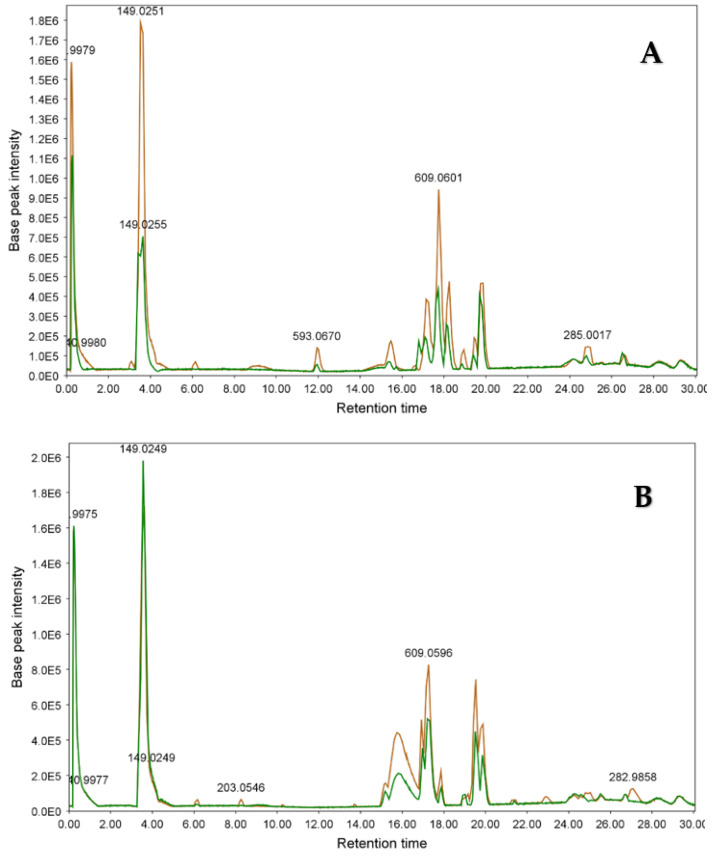
LC-MS chromatograms of 70% (green) and 100% (brown) MeOH extracts of (**A**) *S. alexandrina* and (**B**) *S. italica.* For the separation, XBridge^®^ BEH C18 column (150 × 3.0 mm i.d., particle size 2.5 μm; Waters), gradient elution with MeOH and 0.2% aqueous acetic acid with a flow rate 0.2 cm^3^ min^−1^ were used. E5: 10^5^, E6: 10^6^. (For interpretation of the references to color in this figure legend, the reader is referred to the web version of this article).

**Figure 6 metabolites-13-00559-f006:**
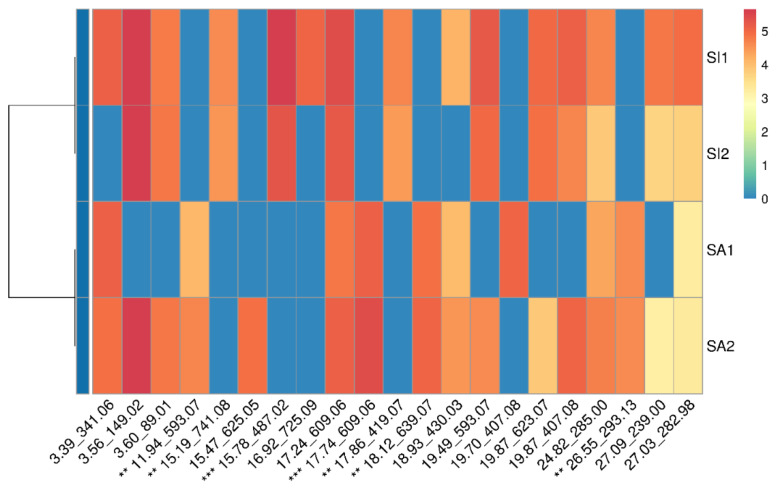
A 2D heat map representing semi-quantitative distribution and metabolite profiling of samples 100% MeOH extract of *S. alexandrina* (SA1), 100% MeOH extract of *S. italica* (SI1), 70% MeOH extract of *S. alexandrina* (SA2), and 70% MeOH extract of *S. italica* (SI2). Sample identity is displayed on the y-axis (right-hand-side of the heatmap). The importance score of each feature is indicated by asterisks. Bayesian clustering was applied. Features (RT and *m*/*z* values) are represented on the x-axis. The color intensity (blue to dark red) refers to compound abundance in each sample. ** *p* < 0.01, *** *p* < 0.001. (For interpretation of the references to color in this figure legend, the reader is referred to the web version of this article).

**Figure 7 metabolites-13-00559-f007:**
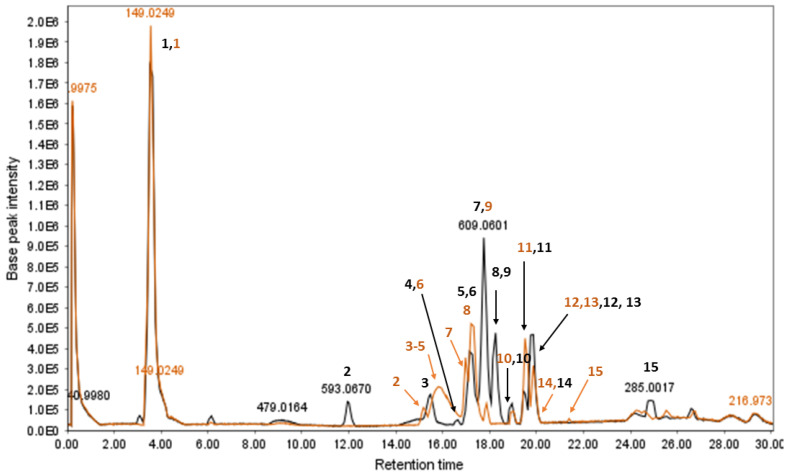
HPLC-ESIMS chromatographic profile of *S. italica* (orange) and *S. alexandrina* (black) 100% MeOH extracts. ESIMS: base peak chromatogram (BPC), negative ion mode, *m*/*z* 150–1500. The compounds were identified by comparing their LC/ESIMS characteristics with literature. Peaks 10s are sennoside A in both samples. Peaks 4 is sennoside B in sample *S. alexandrina*, and peak 6 in sennoside B in *S. italica*. E5: 10^5^, E6: 10^6^. (For interpretation of the references to color in this figure legend, the reader is referred to the web version of this article).

**Figure 8 metabolites-13-00559-f008:**
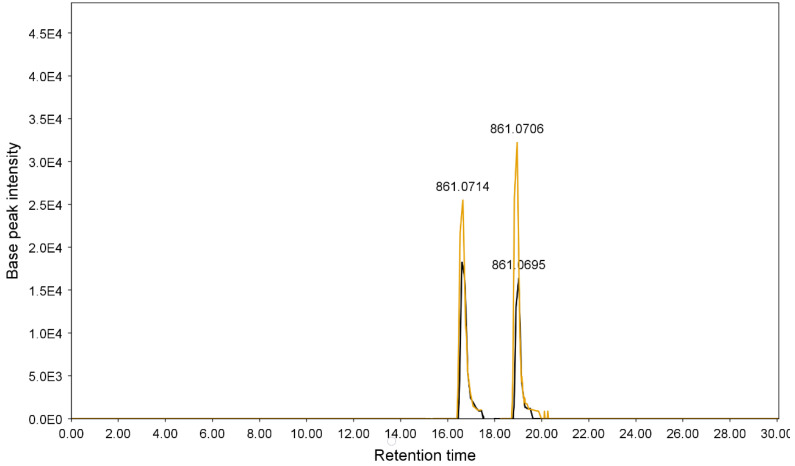
Extracted ion chromatograms (XICs) of sennosides A and B for *S. italica* (black) and *S. alexandrina* (orange) 100% MeOH extracts. ESIMS: extracted ion chromatograms (XIC), negative ion mode, *m*/*z* 861.000–862.0000. E5: 10^5^, E6: 10^6^. (For interpretation of the references to color in this figure legend, the reader is referred to the web version of this article).

**Figure 9 metabolites-13-00559-f009:**
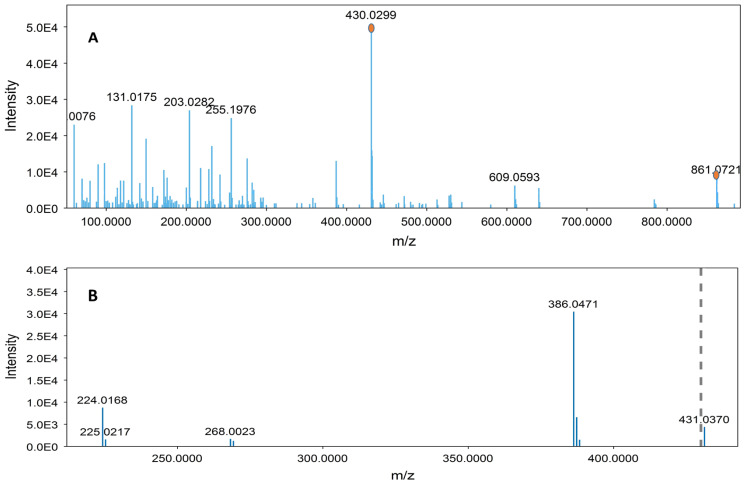
ESI^−^ mass spectra of a sennoside: (**A**) parent [M-H]^−^ ion of *m*/*z* = 861.0721; (**B**) MS2 spectrum of the ion *m*/*z* = 430.0370.

**Figure 10 metabolites-13-00559-f010:**
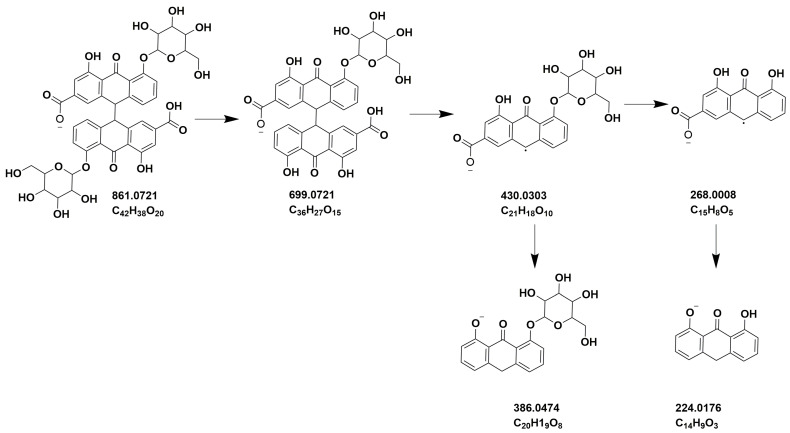
Proposed fragmentation pathway for sennoside A and sennoside B.

**Table 1 metabolites-13-00559-t001:** The botanical characteristics and distribution of *Senna italica* and *Senna alexandrina* [5].

	*Senna italica*	*Senna alexandrina*
Plant	an annual or perennial erect to prostrate herb or subshrub	a subshrub, shrub, or tree
Distribution	seasonally dry tropical biome	subtropical biome
Leaves	3–12 cm long, eglandular	5–15 cm long, eglandular
Stipules	triangular to ovate-triangular, 3–9 mm long	subulate, linear or narrowly triangular, 3–5 mm long
Leaflets	4–6 pairs, obovate-elliptic to obovate-oblong, minutely appressed puberulous	4–8 pairs, lanceolate to elliptic, appressed puberulous or pubescent
Racemes	2–25 cm long	5–30 cm long
Bracts	broadly ovate or elliptic, shortly acuminate, 3–5 mm long, caducous	elliptic to obovate, obtuse to shortly acuminate, 5–11 mm long
Sepals	usually blackish or brownish except for hyaline margins	greenish or hyaline
Petals	yellowish-white to bright yellow, 0.8–2 cm long	yellow or orange-yellow, 0.7–1.7 cm long
Stamens	arranged, 9–10: 2 anthers large, 4–5 medium-sized, 3 small	10: 2 anthers large, 5 medium-sized, 3 small
Pods	shortly oblong-falcate, flattened, 3–6 × 1.3–2 cm, transversely venose, with a ridge of raised crests along middle of each valve	shortly oblong, slightly curved to almost straight, flattened, dehiscent, transversely septate within
Seeds	transversely arranged, compressed, oblong-ovate, apiculate near hilum, often emarginate at opposite end, reticulate, with a small areole on each face	transversely arranged, compressed, oblong or oblong-ovate, apiculate near hilum, often emarginate at opposite end, reticulate or rugose, with a small areole on each face
Native range	Algeria, Angola, Botswana, Burkina, Cameroon, Cape Provinces, Cape Verde, Central African Republic, Chad, Djibouti, Egypt, Eritrea, Ethiopia, Free State, Gambia, Gulf States, India, Iran, Iraq, Kenya, KwaZulu-Natal, Lebanon–Syria, Libya, Mali, Mauritania, Morocco, Mozambique, Namibia, Niger, Nigeria, Northern Provinces, Oman, Pakistan, Palestine, Saudi Arabia, Senegal, Sinai, Somalia, Sudan, Swaziland, Tanzania, Uganda, West Himalaya, Western Sahara, Yemen, and Zimbabwe	Algeria, Central African Republic, Chad, Djibouti, Egypt, Eritrea, Ethiopia, Gulf of Guinea Island, India, Kenya, Mali, Mauritania, Niger, Nigeria, Oman, Pakistan, Palestine, Saudi Arabia, Sinai, Socotra, Somalia, Sri Lanka, Sudan, and Yemen

**Table 2 metabolites-13-00559-t002:** The amounts of sennosides A and B in 70% and 100% MeOH extracts of *S. italica* and *S. alexandrina*. SA1, SA2, SI1, and SI2 refer to *S. alexandrina* 100%, *S. alexandrina* 70%, *S. italica* 100%, and *S. italica* 70% MeOH extracts, respectively.

Sample	Content %	
	Sennoside A ^1^	Sennoside B
SA1	1.85 ± 0.095	0.41 ± 0.12
SA2	1.61 ± 0.38	0.24 ± 0.13
SI1	1.00 ± 0.38	0.32 ± 0.17
SI2	0.61 ± 0.34	0.18 ± 0.14

^1^ The sennosides A and B were measured by LC-MS technique using authentic standards.

**Table 3 metabolites-13-00559-t003:** Identification of the compounds in the chromatograms of *Senna alexandrina* (peak number in Figure 7, retention time in the chromatogram, *m*/*z* of [M-H]^−^ ion, molecular formula, identity, and reference to mass spectrum used for confirmation of the substance).

Peak No.	*t_r_* ^1^ (min)	MS^−^	Formula	MS/MS	Identification	Ref.
**1**	**3.52 ^2^**	**149.0249**	**-**	**89.0131**	**Unresolved**	**-**
2	12.01	593.0678	-	503.0483, 473.0416, 383.0238, 353.0172	Unresolved	-
3	15.47	625.0513	C_27_H_30_O_17_	300.9938	1. Quercetin-O-di-hexoside 2. Hydroxykaempferol di-hexoside	[50]
**4**	**16.61**	**861.0703**	**C_42_H_38_O_20_**	**699.0378, 386.0472, 224.0159, 430.0312**	**Sennoside B**	**[45]**
**5**	**17.17**	**609.0591**	**C_27_H_30_O_16_**	**300.9933**	**Rutin**	**[45,50]**
6	17.32	463.0223	C_21_H_20_O_12_	299.9863, 300.9929	Isoquercitrin/hyperoside	[51]
7	17.75	609.0586	C_27_H_30_O_16_	285.0010	Kaempferol-O-di-glycoside	[45]
8	18.25	639.0654	C_28_H_32_O_17_	315.0075	Isorhamnetin-O-di-hexoside	[50]
9	18.42	529.0751	C_30_H_26_O_9_	289.0320	Cassiaflavan-epicatechin	[52]
**10**	**19.02**	**861.0729**	**C_42_H_38_O_20_**	**430.0303, 386.0474, 224.0176**	**Sennoside A**	**[45]**
**11**	**19.44**	**593.0671**	**C_27_H_30_O_15_**	**447.0313, 285.0010**	**Kaempferol-O-hexoside-pentoside**	**[45]**
**12**	**19.65**	**447.0308**	**C_21_H_20_O_11_**	**283.9939**	**2-Hydroxyemodin glucoside**	**[50]**
**13**	**19.86**	**407.0770**	**C_20_H_24_O_9_**	**245.0477**	**Tinnevellin-O-glucoside** **Torachrysone/Isotorachrysone O-glucoside**	**[45,50]**
14	20.0	623.0706	C_28_H_32_O_16_	315.0070	1. Rhamnetin/isorhamnetin 3-neohesperidoside 2. Nepetin di-hexoside 3. 2′,3′,4′,6,7-Pentahydroxyflavone di-hexoside	[53]
15	24.94	245.0484	C_14_H_14_O_4_	230.0267, 215.0054	Torachrysone/isotorachrysone/tinnevellin	[50]

^1^ Retention time; ^2^ the common compounds in both species are shown in bold.

**Table 4 metabolites-13-00559-t004:** Identification of the compounds in the chromatograms of *Senna italica* (peak number in Figure 7, retention time in the chromatogram, *m*/*z* of [M–H]^−^ ion, molecular formula, identity, and reference to mass spectrum used for confirmation of the substance).

Peak No.	*t_r_*^1^ (min)	MS^−^	Formula	MS/MS	Identification	Ref.
**1**	**3.52 ^2^**	**149.0249**	**-**	**89.0131**	**Unresolved**	**-**
2	15.16	741.0855	-	299.9895	Unresolved	-
3	15.53	581.1065	C_26_H_30_O_15_	563.0961, 257.0457, 239.0385	Norrubrofusarin gentiobioside	[54]
4	15.80	487.0218	-	240.9692	Unresolved	-
5	15.85	431.0377	C_21_H_20_O_10_	269.0079 240.0096	Aloe-emodin glucoside	[50]
**6**	**16.61**	**861.0293**	**C_42_H_38_O_20_**	**430.0293, 386.0465, 224.0171**	**Sennoside B**	**[45]**
7	16.96	445.0155	C_21_H_18_O_11_	282.9854, 239.0021	Cassic acid (rhein) glucoside	[50]
**8**	**17.17**	**609.0604**	**C_27_H_30_O_16_**	**300.9929**	**Rutin**	**[45,50]**
9	17.75	419.0768	C_20_H_20_O_10_	401.0677, 257.0465, 239.0390	De-methyl-toralactone hexoside	[54]
**10**	**19.02**	**861.0695**	**C_42_H_38_O_20_**	**430.0298, 386.0461, 224.0164**	**Sennoside A**	**[45]**
**11**	**19.44**	**593.0665**	**C_27_H_30_O_15_**	**447.0316, 285.0000**	**Kaempferol-O-hexoside-pentoside**	**[45]**
**12**	**19.65**	**447.0301**	**C_21_H_20_O_11_**	**283.9939**	**2-Hydroxyemodin glucoside**	**[50]**
**13**	**19.86**	**407.0768**	**C_20_H_24_O_9_**	**245.0476**	**Tinnevellin-O-glucoside** **Torachrysone/Isotorachrysone O-glucoside**	**[45,50]**
14	20.0	623.0743	C_28_H_32_O_16_	315.0068	1. Rhamnetin/isorhamnetin 3-neohesperidoside 2. Nepetin di-hexoside 3. 2’,3’,4’,6,7-Pentahydroxyflavone di-hexoside	[53]
15	21.38	245.0476	C_14_H_14_O_4_	230.0260, 215.0048	Torachrysone/isotorachrysone/tinnevellin	[45,50]

^1^ Retention time; ^2^ the common compounds in both species are shown in bold.

## Data Availability

Data is contained within the article.

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
