# Peer review of "Comparative LC-ESIMS-Based Metabolite Profiling of Senna italica with Senna alexandrina and Evaluating Their Hepatotoxicity"

_metabolites, 2023, doi:10.3390/metabo13040559_

Round 1

Reviewer 1 Report

In this study, a LCESI/MS-based approach for metabolite profiling was carried out to compare the metabolites of S. italica with S. alexandrina. The cytotoxicity of both species was evaluated against HepG2 cancer cell lines using HPLC-based activity profiling to localize the cytotoxic components and evaluate their safety of use. The research has a certain significance. However, the orderliness of article content needs to be improved. There are several questions for authors to answer.

1. The keyword of “Fabaceae” is broad. Consider removing or replacing it with other word that enable to summarize the main points of the manuscript.

2. For abbreviations appearing in the manuscript, it should be noted behind the first full name in the manuscript, such as: Senna alexandrina (S. alexandrina).

3. The figures and tables should be expressed consistently in the manuscript. Such as: Figure 7, Figure 4 or Fig.7, Fig.4. Please check the full manuscript.

4. Please reorder the result parts and set the appropriate title for each section of the result.

5. The time of the activity profiles in Fig. 1 is not consistent with line 241.

6. The peak of Sennosides A in Fig. 5 is not obvious. Please provide TICs of Sennosides A and B.

7. Whether Sennosides A and B are the main active ingredients for the laxative action. In lines 396-398, for the view that S. italica contains less amount of sennoside A and B, it could also be used as a laxative agent with milder effects, whether there is experimental or other supporting material.

Author Response

Dear Reviewer,

Thank you for giving us the opportunity to submit a revised draft of the manuscript “Comparative LC-ESIMS-based metabolite profiling of Senna italica with Senna alexandrina and evaluating their hepatotoxicity” for publication in the journal of Metabolites. We appreciate the time and effort that you and the reviewers dedicated to providing feedback on our manuscript and are grateful for the insightful comments on and valuable improvements to our paper. We have incorporated all of the suggestions made by the reviewers. The changes are highlighted within the manuscript. Please see below for a point-by-point response to the reviewers’ comments and concerns.

Reviewers' Comments to the Authors

Reviewer 1:

In this study, a LCESI/MS-based approach for metabolite profiling was carried out to compare the metabolites of S. italica with S. alexandrina. The cytotoxicity of both species was evaluated against HepG2 cancer cell lines using HPLC-based activity profiling to localize the cytotoxic components and evaluate their safety of use. The research has a certain significance. However, the orderliness of article content needs to be improved. There are several questions for authors to answer.

  1. The keyword of “Fabaceae” is broad. Consider removing or replacing it with other word that enable to summarize the main points of the manuscript.

Response: Removed.

  1. For abbreviations appearing in the manuscript, it should be noted behind the first full name in the manuscript, such as: Senna alexandrina (Salexandrina).

Response: Corrected accordingly.

  1. The figures and tables should be expressed consistently in the manuscript. Such as: Figure 7, Figure 4 or Fig.7, Fig.4. Please check the full manuscript.

Response: Corrected accordingly.

  1. Please reorder the result parts and set the appropriate title for each section of the result.

Response: Corrected accordingly.

  1. The time of the activity profiles in Fig. 1 is not consistent with line 241.

Response: Thank you for the comment. Corrected accordingly.

  1. The peak of Sennosides A in Fig. 5 is not obvious. Please provide TICs of Sennosides A and B.

Response:  The XICs of sennosides A and B for S. italica and S. alexandrina were added to the manuscript (Figure 7). Besides, in Figure 6, peaks 10s are sennoside A in both samples. Peaks 4 is sennoside B in sample S. alexandrina and peak 6 in sennoside B in S. italica.

  1. Whether Sennosides A and B are the main active ingredients for the laxative action. In lines 396-398, for the view that S. italicacontains less amount of sennoside A and B, it could also be used as a laxative agent with milder effects, whether there is experimental or other supporting material.

Response: thank you for this pointing out. Studies show that sennosides are responsible for the laxative effects of senna. Appropriate references are added and well discussed (Discussion section).

Reviewer 2 Report

Dear Authors

Some major concerns have to fixed before processing with the paper.

·         Abstract:

1.       L26-27: By this, we will be able to examine the feasibility of using Senna italica instead of Senna alexandrina.????? Its not clear you intend to select one of them, why not both ?? please explain

2.       Taken together, according to the results, S. italica can be considered as a substitute for S. alexandrina. How you can decide, please explain ???????????????

·         Introduction

1.       One of your aims was to study the hepatotoxicity of S. italica and S. alexandrina using both in an in vitro model of study: Nothing about this section in the title, abstract, keywords?? Please verify

2.       Please a botanical description of both plant species used in this work

3.       All plant species names have to be in italics and fully written when reported in the first time in the manuscript then Use abbreviation.

·         Methods

1.       L 116: …………identified by botanist Mitra Suzani. Please add rank and affiliation

2.       Please add a figure showing the main differences between the two described plant species

3.       L 126-127: The extracts were stored in a -20 °C freezer after evaporating the solvents. Please add details about how you have evaporated your extract?  Preparation of plant extract: add a reference

·         Results

1.       Please verify the number of tables and figure in the manuscript

2.       Figure 7: please modify, plant species name has to be in italics. Add statistical analysis (mean comparison), add axis, delete cell viability (HepG2).

3.       Figure 1 a and B are not clear. Please modify the axis numbers (have to be same like Y axis)

·         Discussion

Discussion is too basic and superficial. Authors should correlate between the chemical composition of both Senna plant species and the reported biological activity.

Discuss the effect of the external factors affecting the chemical composition of the your plant species selected?

Do you think that only one plant sample material from each plant species is enough to conclude that S. italica can be considered as a substitute for S. alexandrina?? Please explain

·         Conclusion

1.       Please add

Author Response

Dear Reviewer,

Thank you for giving us the opportunity to submit a revised draft of the manuscript “Comparative LC-ESIMS-based metabolite profiling of Senna italica with Senna alexandrina and evaluating their hepatotoxicity” for publication in the journal of Metabolites. We appreciate the time and effort that you and the reviewers dedicated to providing feedback on our manuscript and are grateful for the insightful comments on and valuable improvements to our paper. We have incorporated all of the suggestions made by the reviewers. The changes are highlighted within the manuscript. Please see below for a point-by-point response to the reviewers’ comments and concerns.

Reviewers' Comments to the Authors

Reviewer 2:

Dear Authors

Some major concerns have to fixed before processing with the paper.

  • Abstract:
  1. L26-27: By this, we will be able to examine the feasibility of using Senna italica instead of Senna alexandrina.????? Its not clear you intend to select one of them, why not both ?? please explain.

Response: Thank you very much for the comment. The sentence is revised:" By this, we will be able to examine the feasibility of using Senna italica as a laxative agent like S. alexandrina."

  1. Taken together, according to the results, S. italicacan be considered as a substitute for S. alexandrina. How you can decide, please explain ???????????????

Response: Thank you for pointing this important comment out. The sentence is revised as: "Taken together, according to the results, the metabolite profiles of S. italica and S. alexandrina showed many compounds in common. However, further phytochemical, pharmacological, and clinical studies are necessary to examine the efficacy and safety of S. italica as a laxative agent.”

  • Introduction
  1. One of your aims was to study the hepatotoxicity of S. italicaand S. alexandrina using both in an in vitro model of study: Nothing about this section in the title, abstract, keywords?? Please verify.

Response: The title is changed to: " Comparative LC-ESIMS-based metabolite profiling of Senna italica with Senna alexandrina and evaluating their hepatotoxicity". In addition, all words “cytotoxic” were substituted with “hepatotoxic” in the abstract and keywords.

  1. Please a botanical description of both plant species used in this work.

Response: A table describing the botanical characteristics of both plants were added (table 1).

  1. All plant species names have to be in italics and fully written when reported in the first time in the manuscript then Use abbreviation.

Response: Corrected.

  • Methods
  1. L 116: …………identified by botanist Mitra Suzani. Please add rank and affiliation.

Response: Corrected accordingly: (Department of Pharmacognosy, School of Pharmacy, Mashhad University of Medical Sciences, Mashhad, Iran).

  1. Please add a figure showing the main differences between the two described plant species.

Reply: A figure (Figure 1) showing the plant species was added.

  1. L 126-127: The extracts were stored in a -20 °C freezer after evaporating the solvents. Please add details about how you have evaporated your extract?  Preparation of plant extract: add a reference.

Response: Thank you for the comment. The percolation was used for extraction. The solvent was evaporated by a rotary evaporator. In addition, a reference was added for extraction.

  • Results
  1. Please verify the number of tables and figure in the manuscript

Response: Ten figures and four tables.

  1. Figure 1: please modify, plant species name has to be in italics. Add statistical analysis (mean comparison), add axis, delete cell viability (HepG2).

Response: Thank you for this important comment. Corrected accordingly.

  1. Figure 2 a and B are not clear. Please modify the axis numbers (have to be same like Y axis)

Response: Corrected.

  • Discussion
  1. Discussion is too basic and superficial. Authors should correlate between the chemical composition of both Senna plant species and the reported biological activity.

Response: Some new paragraphs discussing the correlation of laxative effects and metabolite profiles were added.

  1. Discuss the effect of the external factors affecting the chemical composition of your plant species selected?

Response: According to the literature, different factors like the collection season of plants, geographical conditions, etc. might affect the chemical composition of plants. As the aerial parts of senna including the leaflets and fruits have medicinal use, we collected S. italica in the fruiting season of the plant.  

  1. Do you think that only one plant sample material from each plant species is enough to conclude that S. italicacan be considered as a substitute for S. alexandrina?? Please explain.

Response: Thank you for pointing this important comment out. We agree with the reviewer that one sample from each plant might not be enough for deciding the chemical profile of a plant. But our study was a preliminary investigation. Comparing the chemical profile of S. italica from different inhabitat of the plant (different parts of the world) can be the next study to evaluate the chemical profile of a plant species with a higher level of confidence.

  • Conclusion
  1. Please add

Response: Added.

Reviewer 3 Report

The current MS can not be published in its current form. The raised issues below should be carefully addressed.

- First and foremost, the paper needs improving in English language and stylish; a lot of typos can be recognized throughout the text along with repetitions. Punctuation should be improved.

-Abstract should be improved to add more about the results of this work. No need for large introductory sentences in the abstract.

-The type of cytotoxic assay and the results of both extract and control should be added.

-In the plant materials, the identification of Senna alexandrina should be added.

-Why the authors used for extraction 100% MeOH, then 70%, why they didn't start with 70%, clarify this issue, and references should be added.

-The total yield of each plant extract should be added.

_" The file was then loaded into the online web application... " the name of the application should be added.

-The plant's names should be italicized in the MS and figures.

_ Statistical significance, as a horizontal line with asterisks, must be added above the bars.

Why the control was not included in the graph of cytotoxicity with the extracts?

-The Figures started with figure 7, where are the other figures before 7.

-Under each table clarification for each symbol should be added, Δ/ppm

- After the references section, there is a part that does not belong to the MS, and should be removed.

-Conclusion is missing.

-The significance and the impact of this work should be added.

Author Response

Dear Reviewer,

Thank you for giving us the opportunity to submit a revised draft of the manuscript “Comparative LC-ESIMS-based metabolite profiling of Senna italica with Senna alexandrina and evaluating their hepatotoxicity” for publication in the journal of Metabolites. We appreciate the time and effort that you and the reviewers dedicated to providing feedback on our manuscript and are grateful for the insightful comments on and valuable improvements to our paper. We have incorporated all of the suggestions made by the reviewers. The changes are highlighted within the manuscript. Please see below for a point-by-point response to the reviewers’ comments and concerns.

Reviewers' Comments to the Authors

Reviewer 3:

The current MS cannot be published in its current form. The raised issues below should be carefully addressed.

1- First and foremost, the paper needs improving in English language and stylish; a lot of typos can be recognized throughout the text along with repetitions. Punctuation should be improved.

Response: Revised.

2- Abstract should be improved to add more about the results of this work. No need for large introductory sentences in the abstract.

Response: Corrected.

3-The type of cytotoxic assay and the results of both extract and control should be added.

Response: AlamarBlue® assay was used for evaluating the cytotoxic activity. The results for both extract and control added.

4-In the plant materials, the identification of Senna alexandrina should be added.

Response: Corrected. The plant was identified by the botanist Mitra Souzani.

5-Why the authors used for extraction 100% MeOH, then 70%, why they didn't start with 70%, clarify this issue, and references should be added.

Response: Thank you for the comment. To find the best extraction solvent composition, we used 70% and 100% MeOH. The results showed that the yield of extraction and the content of compounds like sennosides A and B were higher in 100% MeOH.

6-The total yield of each plant extract should be added.

Response: Added.

7-" The file was then loaded into the online web application... " the name of the application should be added.

Response: Corrected.

8-The plant's names should be italicized in the MS and figures.

Response: Corrected.

9- Statistical significance, as a horizontal line with asterisks, must be added above the bars.

Response: Corrected.

10-Why the control was not included in the graph of cytotoxicity with the extracts?

Response: Added.

11-The Figures started with figure 7, where are the other figures before 7.

Response: Corrected.

12-Under each table clarification for each symbol should be added, Δ/ppm

Response: Added. Δ/ppm is mass error in ppm.

13- After the references section, there is a part that does not belong to the MS, and should be removed.

Response: Corrected.

14-Conclusion is missing.

 Response: Added.

15-The significance and the impact of this work should be added.

Response: Added to the conclusion.

Round 2

Reviewer 1 Report

The language should be polished.

Reviewer 2 Report

Dear authors

Special thanks and good luck

Reviewer 3 Report

No comment